# Production of a High-Phosphatidylserine Lecithin That Synergistically Inhibits Lipid Oxidation with α-Tocopherol in Oil-in-Water Emulsions

**DOI:** 10.3390/foods11071014

**Published:** 2022-03-30

**Authors:** Harshika Arora, Mitch D. Culler, Eric A. Decker

**Affiliations:** Department of Food Science, University of Massachusetts, Amherst, MA 01002, USA; harshikaaror@umass.edu (H.A.); mitchell.culler@gmail.com (M.D.C.)

**Keywords:** oil-in-water emulsion, α-tocopherol, lecithin, antioxidant, phospholipase D, phosphatidylserine, lipid oxidation

## Abstract

Phosphatidylserine (PS) was shown to work synergistically with tocopherols to extend the shelf life of oil-in-water emulsions. However, the high cost of PS prevents it from being used as a food additive. This work investigated the potential use of a high-PS enzyme-modified lecithin to be used along with α-tocopherol to extend the lag phase of oil-in-water emulsions stabilized using Tween 20. Phospholipase D from *Streptomyces* sp. and L-serine were used to modify lecithin to increase the PS concentration. Enzyme activity was optimized as a function of pH and temperature using high-phosphatidylcholine (PC) soybean, sunflower, or egg lecithins. Under optimal conditions, the final PS concentrations were 92.0 ± 0.01%, 88.0 ± 0.01%, and 63.0 ± 0.02% for high-PC soybean, sunflower, and egg lecithins, respectively. α-Tocopherol (3.0 µmol/kg emulsion) alone increased the lag phase of hydroperoxide and hexanal lag phases by 3 and 4 days compared to the control. Phospholipase-D-modified high-PS soy lecithin increased hydroperoxide and hexanal lag phases by 3 and 4 days, respectively. The addition of phospholipase-D-modified high-PS sunflower and egg lecithin did not have any considerable effects on lag phases compared to the control. The combination of phospholipase-D-modified high-PS lecithins (15.0 µmol/kg emulsion) and α-tocopherol (3.0 µmol/kg emulsion) increased the antioxidant activity of α-tocopherol, increasing the hydroperoxide and hexanal lag phase by 6 and 9 days for soy, 5 and 7 days for sunflower, and 4 and 6 days for egg lecithin, respectively. All phospholipase-D-modified high-PS lecithin–tocopherol combinations resulted in synergistic antioxidant activity (interaction index > 1.0), except for α-tocopherol and high-PS egg lecithin, which showed an additive effect. This research showed that the combination of enzyme-modified high-PS lecithin and α-tocopherol could be an effective and commercially viable clean label antioxidant strategy to control lipid oxidation in emulsions.

## 1. Introduction

A major portion of foods that we consume is present in the form of a heterogeneous mixture of water and oils, which contributes to the taste, texture, and mouthfeel. Nowadays, consumers are actively looking for food products with an ingredient list consisting of known natural ingredients and cleaner labels. With such attention being paid to labels, it has become a task for food industries to fulfill these demands; therefore, research on more natural and healthy food additives has increased. Lipid oxidation remains a challenge in foods because it reduces shelf life due to the formation of off-flavors and -aromas and reaction products that are potentially toxic [1,2,3,4]. In the past, synthetic antioxidants, such as butylated hydroxytoluene (BHT) and tertiary butylhydroquinone (TBHQ), were widely used by industries to control lipid oxidation [5,6]. Clean label strategies to control lipid oxidation include using natural antioxidants, such as tocopherols, as well as rosemary and green tea extracts to replace the synthetic antioxidants. However, these antioxidants are often less effective and more expensive than synthetic antioxidants [7]. Antioxidant combinations are often used to increase the shelf-life of foods that are susceptible to lipid oxidation. This can be more effective than individual antioxidants because the multiple antioxidants can work against different oxidation pathways (e.g., free radical scavenging, metal chelating, and singlet oxygen), partitioning into different phases of the food or via the ability of one antioxidant to regenerate another [5,8,9].

Tocopherols are naturally occurring lipophilic antioxidants with nutritional and health benefits, as well as the potential to be effective food antioxidants. They occur in four different forms, namely, α-, β-, γ-, and δ-tocopherol, which differ in the position and degree of methylation, with α-tocopherol being the most non-polar form [10]. The antioxidant activity of tocopherols primarily comes from their ability to scavenge free radicals [11,12,13]. In some cases, they also act as singlet oxygen quenchers [14,15,16]. Each tocopherol molecule can react with two peroxyl radicals and, thus, block the lipid oxidation chain propagation reactions [17]. Tocopherols initially react with free radicals by donating their phenolic H atom to form a tocopherol radical. The tocopherol radical can then react with another fatty acid radical to scavenge a second radical [18]. The ability of tocopherols to control oxidation is eventually lost when it can no longer scavenge free radicals. An effective strategy to combat lipid oxidation in foods would be to convert the oxidized tocopherols products back to their original structure so that tocopherol can inactivate additional free radicals. Several studies demonstrated the regeneration of oxidized tocopherols by compounds, such as ascorbates, phenols, and phospholipids. For example, Zhou et al. [19] showed that green tea polyphenols were able to regenerate oxidized α-tocopherol back to α-tocopherol in the presence of SDS micelles. Niki et al. [20] studied the combined effect of vitamin E and vitamin C in methyl linoleate, where vitamin C was involved in the regeneration of tocopherol radicals, showing that the two vitamins act synergistically to inhibit lipid oxidation.

Phospholipids are amphiphilic compounds in which the phosphate-containing polar head group forms the hydrophilic end while the hydrophobic end consists of two non-polar fatty acids [21]. Several scientists have reported that phospholipids are able to improve the antioxidant activity of tocopherols and their homologs [22,23,24,25]. Doert et al. [24] showed that the amino-containing phospholipids, namely, phosphatidylethanolamine (PE) and phosphatidylserine (PS), have the potential to regenerate oxidized tocopherols and proposed a reaction scheme for this regeneration by using mass spectroscopy to analyze intermediate reaction products. Cui et al. [25] studied the interaction between phosphatidylcholine (PC) and phosphatidylethanolamine (PE) with α-tocopherol in bulk oil and found that phosphatidylethanolamine (PE) was able to increase the activity of α-tocopherol while phosphatidylcholine did not. They also showed that dioleoylphosphatidylethaolamine (DOPE) was able to regenerate α-tocopherol quinone to α-tocopherol. Samdani et al. [26] showed that PE and PS in combination with tocopherols in oil-in-water emulsions are extremely effective antioxidants, with PS being more effective, as the interaction index between PS and α-tocopherol was 1.5–3 times higher than PE and α-tocopherol when Tween 20 was used as an emulsifier. Xu and coworkers [27] found that PE was more effective than PS in bulk oils, with the highest activity seen in the case of PE and mixed tocopherols. While these studies show that tocopherol plus PE or PS have excellent potential as food additives, they are not currently used in foods because high purity PS and PE are too expensive for use as food additives and both PE and PS concentrations in commercial lecithins are not high enough to strongly increase the activity of tocopherols.

Lecithin is a commonly used ingredient in the food industry because of its low cost, abundant availability, and functional properties. It contains different phospholipids, such as phosphatidylcholine (PC), phosphatidylethanolamine (PE), and phosphatidylinositol (PI) (Table 1) [28]. Recently, there has been rising interest in modifying the properties of lecithin using lipolytic enzymes, such as phospholipases and lipases. Manipulation in the polar head group composition of native phospholipids using enzymes is gaining interest because it reduces the high cost associated with the industrial purification of phospholipids [29,30]. The enzyme phospholipase D can be used to perform this head group exchange via a process called transphosphatidylation in which the enzyme cleaves the phosphodiester bond to remove the head group and attaches another headgroup available in the surrounding environment to form a new phospholipid structure [29,31,32]. Hence, lecithin can be modified by phospholipase D to a high PS or PE lecithin by converting the PC to PS or PE in the presence of primary alcohols, such as serine (Figure 1) or ethanolamine.

A biphasic system of an organic phase to dissolve the phospholipids and a water/buffer phase consisting of the enzyme and the serine or ethanolamine is traditionally used for transphosphatidylation using phospholipase D. The advantage of this system is that the reactants in the two phases can be easily separated once the target phospholipid is formed [32]. However, a challenge with this system is that phospholipase D often competes between two reaction pathways (transphosphatidylation and hydrolysis). When the conditions favor hydrolysis, the final product is phosphatidic acid, resulting in lower PS or PE yields. Thus, understanding how reaction conditions impact the type of product formed is extremely critical for good yields of the phospholipid of interest.

There are many naturally available phospholipase D sources from plants, but these typically have low transphosphatidylation activity. Therefore, a commercially available microbial phospholipase D enzyme was selected for this study because of its high activity and ability to favor transphosphatidylation over hydrolysis [33]. Most of the published studies showing phospholipase D conversion used pure PC as the substrate instead of commercial lecithin. Besides phospholipids, commercial lecithin contains other lipids, such as free fatty acids, glycolipids, sterols, and other minor components [34], that could affect phospholipase D’s ability to modify PC. Therefore, it is important to optimize the conditions for the production of a high-PS lecithin using phospholipase D and commercial lecithins.

The overall objective of this study was to determine whether phospholipase D could produce a high-PS modified lecithin for use as a food antioxidant. In this study, we focused on the ability of phospholipase D to convert PC to PS in three different types of food-grade lecithins. The antioxidant activity of the phospholipase-modified high-PS lecithins was then tested in oil-in-water emulsions. PS was chosen because it was found to be more effective than PE when used in combination with tocopherols in oil-in-water emulsions [26].

## 2. Material and Methods

### 2.1. Materials

High PC soy lecithin (~94% PC) and high PC sunflower lecithin (~90% PC) were obtained from the American Lecithin Company (Oxford, CT, USA). Egg lecithin (60% PC) was obtained from Alfa Aesar through Thermo Fisher Scientific (Ward Hill, MA, USA). Authentic phospholipids mixed in chloroform (1,2-dioleoyl-sn-glycero-3-phosphocholine (PC), 1,2-dioleoyl-sn-glycero-3-phosphoethanolamine (PE), 1,2-dioleoyl-sn-glycero-3-phospho-L-serine (PS), and 1,2-dioleoyl-sn-glycero-3-phosphate (sodium salt) (PA)) were procured from Avanti Polar Lipids, Inc. (Alabaster, AL, USA) and were stored at −20 °C. Soybean oil was purchased from a local retail store in Hadley, MA, USA, and was stored at −80 °C until use. L-serine, calcium chloride, iso-octane, 2-propanol, methanol, 1-butanol, *n*-hexane, hydrochloric acid, chloroform, and ethyl acetate were supplied by Fisher Scientific (Fair Lawn, NJ, USA). Phospholipase D from *Streptomyces* sp. (500 units/mL), *Streptomyces chromofuscus* (>50,000 units/mL), silicic acid (100–200 mesh), activated charcoal (100–400 mesh), sodium acetate anhydrous, imidazole, Tween 20, barium chloride dihydrate, ammonium thiocyanate, iron (II) sulfate heptahydrate, cumene hydroperoxide, hexanal, and (±)-α-tocopherol were supplied by Sigma-Aldrich (St. Louis, MO, USA). All the solvents were of HPLC grade or purer, and all other chemicals were analytical grade or purer. Double distilled and deionized water was used for all the experiments and all glassware was soaked in a 2 N HCl bath overnight to remove metals, followed by rinsing with double-distilled water before use.

### 2.2. Preparation of Modified Lecithin

The protocol to produce high phosphatidylserine lecithin was developed using a modified version of the enzymatic conversion method reported by Hosokawa et al. [35]. Phospholipase D from *Streptomyces chromofuscus* and *Streptomyces* sp. were both tested for lecithin modification [36,37,38]. A buffer solution was prepared, which consisted of 0.2 M sodium acetate, 0.01 M CaCl_2_, and 2.5 M L-serine. The pH was adjusted as required using HCl. A buffer (6 mL) was mixed with 8 units of phospholipase D enzyme by vortexing for 10 s. Lecithins (0.3 g), mixed with 25 mL of ethyl acetate, were added to the enzyme solution to form a biphasic system. The mixture was incubated at 37 °C for up to 40 h with continuous shaking at 150 rpm (Brunswick Innova 2100 platform shaker, Eppendorf, Hamburg, Germany) to carry out the enzymatic conversion. At various times, the phospholipase D was inactivated by heating at 90 °C for 5 min and ethyl acetate was removed from the samples using a rotary evaporator at 37 °C. The remaining water phase was further combined with 80 mL chloroform, 30 mL methanol, and 20 mL of water in a 250 mL separatory funnel and allowed to rest for at least 60 min to allow for the separation of the solvents and aqueous phase. The organic phase was transferred to a round bottom flask and the solvent was evaporated at 42 °C in a rotary evaporator to obtain the modified phospholipids. The modified lecithin was stored in amber-colored bottles at −80 °C until use.

### 2.3. Determination of Tocopherol Content of the Lecithins

Tocopherols were quantitated by normal phase HPLC [39] using a mixture of n-hexane and 1,4-dioxane (95:5, *v*/*v*) as the mobile phase at a flow rate of 1 mL/min. The lecithins (0.01 g) were dissolved in 2 mL of n-hexane and aliquoted into HPLC vials using a 0.2 µm syringe filter. Each sample (20 μL) was injected into Shimadzu Prominence-i LC-2030C HPLC system equipped with a Supelcosil LC Diol column (L × i.d. = 25 cm × 4.6 mm, 5 µm particle size). The effluent was monitored with a fluorescence detector (Shimadzu RF—20A) with an excitation wavelength at 290 nm and emission wavelength at 330 nm. The tocopherols were identified using retention times and were quantified with standard curves (peak areas) obtained from authentic standards.

### 2.4. Detection of Phospholipids by HPLC

To identify the phospholipids composition of the unmodified and modified lecithin using HPLC, a modified version of the method described by Letter [40] was developed. The samples were analyzed using a Shimazu Prominence-i LC-2030C HPLC (Kyoto, Japan) with an electron light scattering detector ELSD-LT II. Separation was carried out using an Agilent Zorbax SIL, 4.6 mm i.d. × 250 mm column containing 5 μm packing at 35 °C. The mobile phase was a low-pressure tertiary gradient of hexane, isopropanol, and water, as described in Table 2. Authentic phospholipids, unmodified lecithins, and modified lecithins (10 mg) were dissolved in 1.5 mL of 53% hexane, 5% water, and 42% isopropanol. The samples were vortexed and filtered through a 0.2 μm syringe filter and aliquoted into 2 mL autosampler vials of which 10 μL of each sample was injected into the HPLC system for analysis. Compressed nitrogen was used as the carrier gas in the evaporative light scattering detector at a temperature of 63 °C, an internal pressure of 400 kPa, and a gain of 5. Phospholipids were identified and the percentage conversion, i.e., (peak area of individual phospholipid)/(total peak area of detected phospholipids) × 100%, was calculated using the peak areas of each phospholipid.

### 2.5. Preparation of Stripped Soybean Oil (SSO)

Stripped soybean oil was used to decrease the interference of polar lipids including tocopherols and phospholipids in the lipid oxidation studies. Oil stripping was performed as described by Cui et al. [41] using a chromatographic column (3.0 cm internal diameter × 35 cm height), which was packed with 4 layers. The first layer was packed with sand (around 2 cm). The second layer was prepared using 22.5 g of silicic acid (washed three times using double distilled and deionized water and dried for 48 h at 110 °C), followed by a layer of activated charcoal (5.25 g). The top layer was packed using another 22.5 g of silicic acid. Soybean oil (30 mL) was mixed in 30 mL n-hexane and passed through the column using an additional 270 mL of n-hexane for complete elution from the column to obtain the stripped oil. The column was covered with aluminum foil and a round bottom flask was used to collect the solvent with the oil was put on ice to decrease light- and temperature-induced lipid oxidation during the stripping process. The final oil was collected, and n-hexane was removed using a vacuum rotary evaporator (Model RE 111, Buchi, Flawil, Switzerland) at 27 °C, followed by nitrogen flushing for 10 min to remove traces of hexane from the stripped oil. The oil was stored at −80 °C until it was used for emulsion preparation. The absence of tocopherols in SSO was confirmed using HPLC [39].

### 2.6. Emulsion Preparation and Storage Conditions

Tween 20 was dissolved in 10 mM imidazole-acetate buffer at pH 7.0. Stock solutions of authentic PS, phospholipase-D-modified high-PS lecithins, and unmodified lecithin were prepared in chloroform, and α-tocopherol stock solutions were prepared in ethanol. These stock solutions were added to the stripped oil alone or in combination; the addition to the oil was done at 4 °C in the dark for 30 min with continuous magnetic stirring [26]. Stripped oil was mixed with the imidazole-acetate buffer containing Tween 20 (1:10 emulsifier/oil ratio) and was blended at a low speed to make a coarse emulsion using a handheld homogenizer (Model M133/1281-0, BioSpec Products Inc., Bartlesville, OK, USA). The coarse emulsion was then passed through a microfluidizer (Model M-110L Microfluidics, Newton, MA, USA) at 9 kbar for 3 passes to further reduce the particle size. The emulsion and the homogenizer chamber were kept cold at all times using ice to maintain the emulsion at or below room temperature. One milliliter of the emulsion was aliquoted into 10 mL gas chromatography (GC) vials, sealed using metal caps having PTFE/silicone septa, and stored at 20 °C in dark for the storage studies.

### 2.7. Emulsion Droplet Size and Zeta Potential

Particle size and zeta potential measurements were performed immediately after the emulsions were prepared and at the end of the storage studies by diluting the emulsion with a 10 mM acetate-imidazole buffer (pH 7.0) to keep the attenuation within the range of 6–8. Measurements were taken in triplicates using a Zetasizer Nano-ZS (Malvern Instruments, Worcestershire, UK) instrument at room temperature.

### 2.8. Measurement of Primary Oxidation Products

To monitor the formation of primary oxidation products in the presence of the different antioxidants, lipid hydroperoxides were measured using the method developed by Shantha and Decker [42] with some modifications. Hydroperoxide measurements were performed daily after gas chromatography (GC) analysis (below). Emulsions (0.3 mL) were pipetted out of the GC vials and mixed with 1.5 mL of isooctane:2-propanol solution (3:1, *v*/*v*). The mixture was vortexed for 30 s and then centrifuged for 3 min at 3000× *g* (CL10 centrifuge, Thermo Fisher Scientific Inc., Waltham, MA, USA). The top solvent layer (200 μL) was mixed with 2.8 mL of a methanol:1-butanol solution (2:1, *v*/*v*), followed by the addition of 30 µL of a 1:1 mix of ammonium thiocyanate and ferrous iron. The ferrous ion was prepared from a mixture of 0.144 M FeSO_4_ solution with 0.132 M BaCl_2_ (in 0.4 M HCl) at a 1:1 ratio. The mixture was centrifuged for 5 min at 3000× *g* and 1 mL of the clear supernatant was mixed with 1 mL of 3.94 M ammonium thiocyanate via vortexing and then incubated for 20 min in the dark at room temperature. The absorbance of the samples was measured using a UV-Vis spectrophotometer (Genesys 20, Thermo Fisher Scientific Inc., Waltham, MA, USA) at a wavelength of 510 nm. Samples that showed high absorbance values (>1.2) were diluted 10 times with methanol/1-butanol (2:1, *v*/*v*) before the measurements. A standard curve was prepared using different concentrations of cumene hydroperoxide to calculate the final hydroperoxide concentration.

### 2.9. Measurement of Secondary Oxidation Products

Headspace hexanal measurements were performed via solid-phase microextraction coupled to a gas chromatograph with a flame ionization detector (SPME-GC-FID) using a Shimadzu GC-2010 with a Shimadzu AOC-6000 autosampler (Shimadzu, Kyoto, Japan) [43]. GC vials were heated in an autosampler at 55 °C for 10 min. After heating, the volatiles were absorbed on the surface of a 50/30 mm divinylbenzene/carboxen/polydimethylsiloxane SPME fiber needle (Supelco, Bellefonte, PA, USA), which was inserted for 2 min into the GC vials at the end of the heating process. The fiber needle carrying the volatile compounds was placed into the injector port of the GC where they are desorbed at 250 °C for 3 min and separated on a 30 m × 0.32 mm i.d. × 1 μm fused silica capillary Equity-1 column for 10 min using helium as the carrier gas. The oven temperature was 65 °C, while the FID was at 250 °C and a split ratio of 1:7 was used. The area under the curve was used for quantification using a standard curve prepared with (0–200 μM) authentic hexanal.

### 2.10. Interaction Index

The interaction index [43,44] is a parameter used to determine whether two antioxidants have a synergistic, additive, or antagonist relationship. If the interaction index value for the increase in hydroperoxide or hexanal lag phase for the α-tocopherol and lecithin combination is >1 then the relationship is synergistic, additive if the value = 1, and antagonist if the value is <1. The formula used to determine the index value is as follows: (lag phase of the high-PS lecithin and tocopherol combination − lag phase of the control)/[(lag phase of tocopherol alone − lag phase of the control) + (lag phase of the high-PS lecithin alone − lag phase of the control)].

### 2.11. Statistical Analysis

All treatments were prepared in triplicates and data were presented as means ± standard deviation. Data were analyzed using one-way analysis of variance (ANOVA) followed by Dunnett’s post hoc test (*p* < 0.05) using Minitab 20 version (State College, PA, USA). Data obtained from each day were compared against a control (day 0) to determine the lag phases for each treatment. In this study, the lag phase was defined as the first data point that was statistically greater than day 0.

## 3. Results and Discussion

### 3.1. Effect of Enzyme Source

While the conditions for phospholipase D transphosphatidylation of PC to PS were reported using authentic PC [37,38,45], optimal conditions have not been established when lecithin is the substrate. This could be important since lecithin contains components besides phospholipids, such as triacylglycerols, free fatty acids, mono- and diacylglycerols, tocopherols, and sterols, that could impact the enzyme activity. In addition to the formation of PS, we also determined the formation of PA, as this would identify conditions that could favor hydrolysis vs. transphosphatidylation and, thus, could decrease the PS yield.

High PC soybean lecithin was used for all initial experiments, as its high PC concentration had the potential to obtain a high PS yield. Two commercially available phospholipase D sources, namely, *Streptomyces chromofuscus* and *Streptomyces* sp., were tested using the conditions described by Xu et al. and Hosokawa et al. [27,35], respectively. No PS formation was observed in the presence of phospholipase D from *Streptomyces chromofuscus*. This disagrees with the findings of Duan et al. [38], who were able to obtain up to 70% PS formation from purified PC after 12 h of incubation using this enzyme at 40 °C with 2-methyltetrahydrofuran as the solvent. When *Streptomyces* sp. was used under the same conditions as *Streptomyces chromofuscus*, PS formation was observed, although the yield was low. This suggested that the reaction conditions and the use of lecithin instead of authentic PC could have had an impact on the performance of the two enzymes. Therefore, we screened a variety of conditions (pH, temperature, enzyme concentration, serine concentration, buffer volume) using phospholipase D from *Streptomyces* sp. [35,36,37] since it was more active than *Streptomyces chromofuscus*. Each condition was optimized one by one by doing a 24 h incubation study (Table 3). Increasing PLD above eight units increased the formation of PA (hydrolysis), therefore eight units was chosen for the remaining experiments. Serine at 2.5 M was chosen, as precipitation was observed at 3.0 M. A buffer concentration of 6.0 mL was chosen, as higher concentrations increased PA formation. Kinetics studies on the effect of PS formation as a function of pH and temperature were then performed to be able to not only determine the optimal pH and enzyme stability but also to determine the optimal time.

### 3.2. Effect of pH and Temperature on Phospholipase D Transphosphatidylation Activity

Over the pH range of 5.0–6.5, the optimal pH was found to be 5.5 (Figure 2). For example, after 24 h of incubation, PS was 16, 11, and 14% higher at pH 5.5 compared to pH 5.0, 6.0, and 6.5, respectively. This is similar to what was observed by Juneja et al. [36], who used *Streptomyces* sp. enzyme from two different manufacturers and, in both cases, more than 85% PS was obtained using authentic PC as the substrate at pH 5.5 in the presence of 3.4 M serine.

Additionally, PA formation was also the lowest at pH 5.5. Negatively charged surface-active lipids, such as PA, can sometimes be pro-oxidative in foods because they attract transition} metals [46,47]; therefore, minimizing the PA concentration could result in better antioxidant activity.

Phospholipase D conversion of high-PC lecithin to PS and PA at pH 5.5 as a function of temperature is shown in Figure 3. The enzyme showed good thermal stability and was active between 32 °C and 55 °C. PS formation increased from 32 °C to 37 °C and then decreased from 37 °C to 55 °C. Less than 5% PA was formed at 37 °C, while PA formation was higher at 32 °C and 55 °C. Higher PA formation could partially help to explain the lower PS yield at 32 °C and 55 °C. Due to the higher PS and lower PA yield, 37 °C was selected for the preparation of the phospholipase-D-modified high-PS lecithins used in the lipid oxidation studies. Since PS formation continued to increase and PA concentrations remained low at 37 °C throughout the incubation study, 40 h was chosen for the incubation time. The final composition of the high-PS soy lecithin was 92.0 ± 0.01% PS, 5.0 ± 0.01% PC, and 3.0 ± 0.01% PA.

### 3.3. Emulsion Droplet Size and Charge

No major changes in emulsion droplet size were observed during the entire incubation period of the oxidation studies. The average particle size was 183.8 ± 4.3 nm. The samples showed no signs of creaming or phase separation. The zeta potential for the Tween-20-stabilized emulsions averaged −11.4 ± 1.04 mV.

### 3.4. Impact of Unmodified Soybean Lecithin with or without α-Tocopherol on the Oxidation of Oil-in-Water Emulsions

The ability of α-tocopherol and unmodified soybean lecithin to inhibit lipid oxidation in Tween-20-stabilized oil-in-water emulsions was determined by monitoring the headspace hexanal and lipid hydroperoxides formation at pH 7.0. α-Tocopherol was chosen because it is widely used by industries as an additive to control lipid oxidation because it is inexpensive and label-friendly. The concentrations of α-tocopherol (3 µmol/kg of emulsion) and lecithin (15 µmol/kg emulsion) were chosen because these concentrations worked well together, and synergism was observed with an interaction index of >5 in 1% oil-in-water emulsions in the work done by Samdani et al. [26]. In the control (no α-tocopherol or lecithin), the hydroperoxide and hexanal lag phases were 1 and 2 days, respectively (Figure 4). The addition of unmodified lecithin (15 µmol/kg emulsion) had a small effect as it increased the hexanal lag phase by 1 day while the hydroperoxide lag phase was unchanged as compared to the control. When α-tocopherol (3 µmol/kg emulsion) was added alone, the hydroperoxide and hexanal lag phases were extended to 3 days and 4 days, respectively, compared to the control. When unmodified lecithin (15 µmol/kg emulsion) was added with α-tocopherol in the emulsion, the hydroperoxide and hexanal lag phases were unchanged compared to α-tocopherol alone. This data showed that unmodified lecithin had little antioxidant activity on its own and did not increase the antioxidant activity of α-tocopherol.

### 3.5. Ability of Phospholipase-D-Modified High-PS Soy Lecithin and/or α-Tocopherol to Inhibit Lipid Oxidation in Oil-in-Water Emulsions

The low-PS, unmodified lecithin had little antioxidant activity in the oil-in-water emulsion; therefore, further studies were conducted to determine whether increasing the PS concentrations in the lecithin increased the antioxidant activity. The activity of the phospholipase-D-modified high-PS soy lecithin was also compared to authentic PS at equal PS concentrations. In the control (no α-tocopherol or PS source), the hydroperoxide and hexanal lag phases were again 1 and 2 days, respectively. In the presence of authentic PS (15 µmol/kg emulsion), the hydroperoxide and hexanal lag phases were 1 day and 3 days, respectively, indicating that the presence of phospholipid alone had very little effect on the oxidative stability of the emulsion. This was similar to the observations of Samdani and coworkers [26]. Samples containing phospholipase-D-modified high-PS soy lecithin alone showed longer hydroperoxide and hexanal lag phases of 3 days and 4 days, respectively, compared to the control. The higher antioxidant activity of the modified lecithin alone suggested that something else was in the high-PS modified lecithin that was inhibiting lipid oxidation. This was not likely to be PS since the authentic PS alone did not increase the lag phase. One possibility was that the soy lecithin contained small amounts of naturally occurring α-tocopherol (2.0 g/kg lecithin), which is equivalent to 60 nmol α-tocopherol/kg of emulsion. This level of tocopherol was not able to inhibit oxidation by itself since the lag phases of the unmodified lecithin alone were similar to the control (Figure 5). However, in the phospholipase-D-modified high-PS soy lecithin, the combination of the endogenous tocopherols and PS could result in the inhibition of lipid oxidation, as was observed by the increase in both hydroperoxide and hexanal lag phases compared to the control. When α-tocopherol (3 µmol/kg of emulsion) was added alone, the hydroperoxide and hexanal lag phases were extended to 3 days and 4 days, respectively, compared to the control. The addition of authentic PS (15 µmol/kg emulsion) with α-tocopherol (3 µmol/kg emulsion) extended the hydroperoxide lag phase to 6 days and the hexanal lag phase to 9 days, resulting in a synergistic effect with an interaction index of 2.5 for hydroperoxides and 2.3 for hexanal. The combination of the phospholipase-D-modified high-PS soy lecithin (15 µmol PS/kg emulsion) and α-tocopherol at (3 µmol/kg emulsion), the hydroperoxide lag phase was 6 days and hexanal lag phase was 9 days, resulting in an interaction index of 1.25 for hydroperoxides and 1.75 for hexanal, which again showed synergistic behavior. It was somewhat surprising that the phospholipase-D-modified high-PS soy lecithin + α-tocopherol combination did not show greater antioxidant activity than the authentic PS + α-tocopherol combination since the phospholipase-D-modified high-PS soy lecithin alone had greater antioxidant activity than the authentic PS alone. It is not clear why a difference was not observed, but it could have been the case that the amount of tocopherol in the high-PS soy lecithin was not high enough to further increase the antioxidant activity in the presence of 3 µmol α-tocopherol/kg emulsion.

### 3.6. Ability of Sunflower and Egg Lecithin with or without α-Tocopherol to Inhibit Lipid Oxidation in Oil-in-Water Emulsions

The effect of the addition of lecithin (unmodified and phospholipase-D-modified high-PS lecithin) from sunflower and egg to the oil-in-water emulsion with or without α-tocopherol was also analyzed at equivalent PS concentrations to determine whether different sources of lecithin influenced the ability of modified lecithin to act synergistically with α-tocopherol. The final composition of the phospholipase-D-modified high-PS sunflower lecithin was 88.0 ± 0.01% PS, 3.0 ± 0.01% PC, and 8.0 ± 0.01% PA, and the final composition of the phospholipase-D-modified high-PS egg lecithin was 63.0 ± 0.02%, 6.0 ± 0.02% PC, and 29.0 ± 0.02% PA. Since the concentration of PC in egg lecithin was lower than the other sources, a lower PS yield was obtained. Since the PS concentrations were different, the stock solutions of each lecithin were prepared and added to the emulsion at final concentrations of 12.86, 20.06, and 28.33 mg/kg emulsion for soy, sunflower, and egg lecithin, respectively.

The Tween-20-stabilized oil-in-water emulsion control had a lag phase for lipid hydroperoxides formation of 1 day and the lag phase of hexanal formation was 2 days. The addition of α-tocopherol (3.0 µmol/kg emulsion) to the Tween-20-stabilized emulsions extended the hydroperoxide lag phase to 3 days and the hexanal lag phase to 4 days. The addition of unmodified sunflower lecithin alone did not affect the hydroperoxide and hexanal lag phase as compared to the control. This was different from unmodified soy lecithin, even though they both had similar tocopherol concentrations (2.0% for soy vs. 2.4% for sunflower). When the combination of unmodified sunflower lecithin (15.0 µmol/kg emulsion) and α-tocopherol (3.0 µmol/kg emulsion) was added to the emulsion, the combination extended the hydroperoxide lag phase to 3 days and the hexanal lag phase to 4 days, which was similar to the lag phases obtained when α-tocopherol was added alone.

The addition of phospholipase-D-modified high-PS sunflower lecithin (15.0 µmol/kg emulsion) alone to the Tween-20-stabilized emulsions resulted in a hydroperoxide lag phase of 1 day and a slight increase in the hexanal lag phase to 3 days compared to the control (Figure 6). The combination of high-PS sunflower lecithin and α-tocopherol extended the hydroperoxide lag phase to 5 days and the hexanal lag phase to 7 days, resulting in an interaction index of 2 for hydroperoxides and 1.6 for hexanal.

The addition of 15.0 µmol unmodified egg lecithin/kg of emulsion gave hydroperoxide and hexanal lag phases similar to the control, showing that it had no antioxidant activity (Figure 7). When unmodified egg lecithin (15.0 µmol/kg emulsion) was added with α-tocopherol (3.0 µmol/kg emulsion), the lag phases for both hydroperoxides and hexanal were again similar to the lag phases obtained for α-tocopherol alone (3.0 µmol/kg emulsion), indicating no synergism. The addition of 15.0 µmol/kg of phospholipase-D-modified high-PS egg lecithin increased both the hydroperoxide and hexanal lag phase by 1 day compared to the control. When phospholipase-D-modified high-PS egg lecithin (15.0 µmol/kg emulsion) was added with α-tocopherol (3.0 µmol/kg emulsion), the lag phases for both hydroperoxides and hexanal were 4 and 6 days, with interaction indexes of 1 and 1.3 for hydroperoxide and hexanal, respectively.

The lower increase in hydroperoxide and hexanal lag phase of the high-PS egg lecithin compared to the high-PS sunflower and soybean lecithins could have been due to its higher PA concentration (29% vs. 3–8%). In addition, egg lecithin could have contained more transition metals that could promote oxidation. Aleksandrovna et al. [48] reported that egg lecithin contains 1.8 mg iron/kg lecithin and 0.7 mg copper/kg lecithin compared to soy lecithin, which has 0.4 mg iron/kg lecithin and 0.3 mg copper/kg lecithin, and sunflower lecithin, which has 0.6 mg iron/kg lecithin and 0.4 mg copper/kg lecithin. Eggs contain iron and copper for the nutrition of the chick. Since phospholipids are negatively charged, it would not be surprising that they would associate with the transition metals during the lecithin isolation. The addition of the unmodified or modified egg lecithin by itself did not increase oxidation rates, even though the emulsions could contain higher levels of iron and copper. This could occur if the iron and copper in the egg lecithin were in their oxidized, less reactive forms. However, the decrease in lag phases in the presence of tocopherols and the phospholipase-D-modified high-PS egg lecithin could have been due to the ability of tocopherols to reduce the transition metals and promote oxidation in oil-in-water emulsions [49]. Therefore, tocopherol could have been acting as both a prooxidant by reducing metals and an antioxidant by scavenging free radicals (especially in the presence of PS), with the antioxidant activity being stronger thus allowing extensions of the lag phase compared to the control and α-tocopherol alone.

In conclusion, this research showed that utilizing *Streptomyces* sp. phospholipase D to increase the PS concentration of commercial lecithins was a viable method to produce modified lecithin that could increase the antioxidant activity of α-tocopherol in Tween-20-stabilized oil-in-water emulsions. Over 88% PS was formed in both the high-PC soy and sunflower lecithins after conversion, while the egg lecithin, which contained 63% PC, produced 63% PS. At equal PS concentrations, both the modified high-PS soy and sunflower lecithins synergistically increased the antioxidant activity of α-tocopherol, while egg lecithin only produced an additive effect. The soy and sunflower lecithins were able to increase the tocopherol lag phase by 1.6-fold. These data suggested that a high-PS modified lecithin could be an effective tool for increasing the shelf-life of oil-in-water emulsions.

## Figures and Tables

**Figure 1 foods-11-01014-f001:**
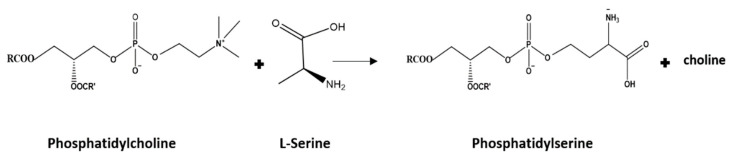
Reaction scheme of transphosphatidylation to PS from PC.

**Figure 2 foods-11-01014-f002:**
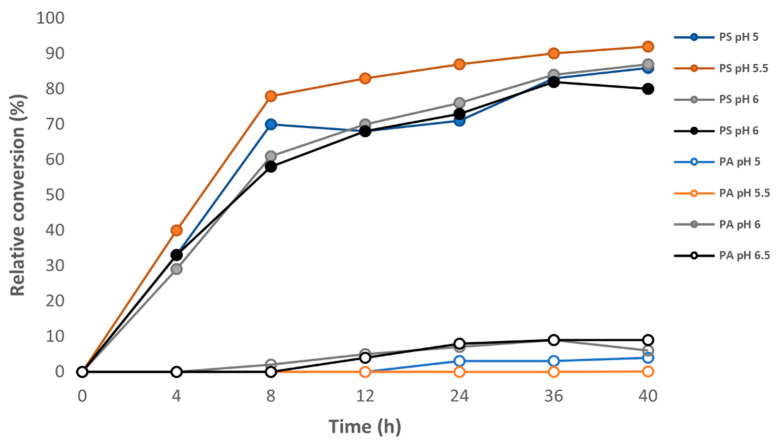
Effect of pH on the formation of phosphatidylserine (PS) from high-phosphatidylcholine soy lecithin. Reaction conditions: 6 mL of 0.2 M acetate buffer containing 2.5 M L-serine, 0.01 M CaCl_2_, 25 mL ethyl acetate, 0.3 g of soy lecithin, 8 units phospholipase D, T = 37 °C. PS—phosphatidylserine (solid symbols), PA—phosphatidic acid (open symbols). Error bars lie within the data points.

**Figure 3 foods-11-01014-f003:**
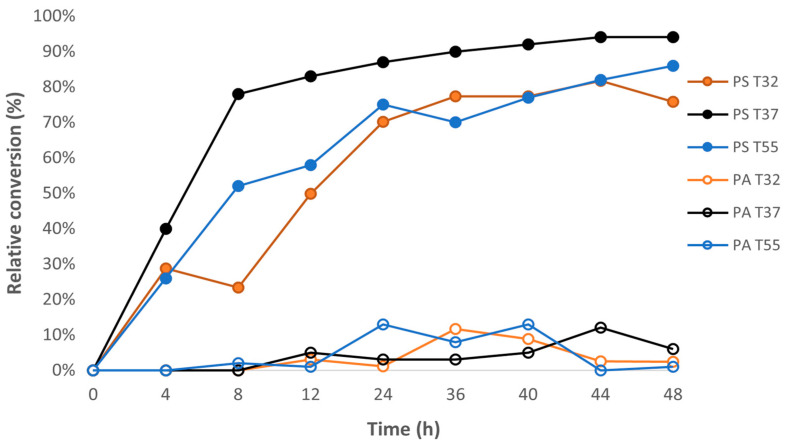
Effect of temperature on the conversion of PC to PS and PA from high-phosphatidylcholine soy lecithin. Reaction conditions: 6 mL of 0.2 M acetate buffer containing 2.5 M L-serine, 0.01 M CaCl_2_, 25 mL ethyl acetate, 0.3 g of soy lecithin, 8 units phospholipase D, pH = 5.5. PS—phosphatidylserine (solid symbols), PA—phosphatidic Acid (open symbols). Error bars lie within the data points.

**Figure 4 foods-11-01014-f004:**
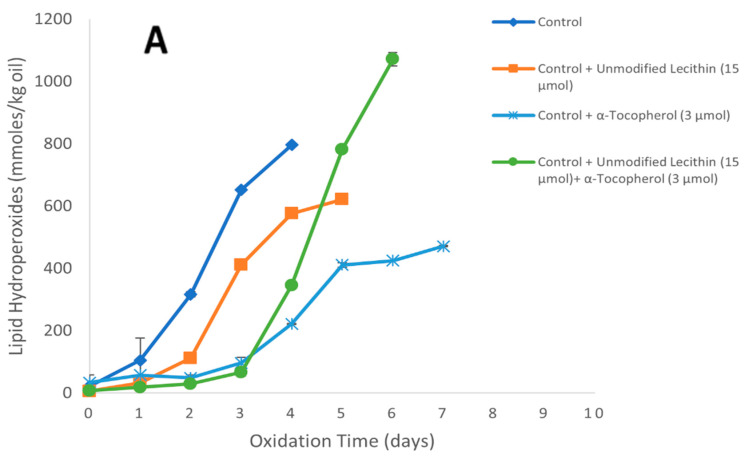
Formation of lipid hydroperoxides (**A**) and hexanal (**B**) in 1% stripped soybean oil-in-water emulsions stabilized with Tween 20 containing 3.0 µmol α-tocopherol/kg of emulsion and/or 15.0 µmol unmodified lecithin/kg emulsion at 20 °C. Each value represents the mean (*n* = 3) ± standard deviations. Some error bars lie within the data points.

**Figure 5 foods-11-01014-f005:**
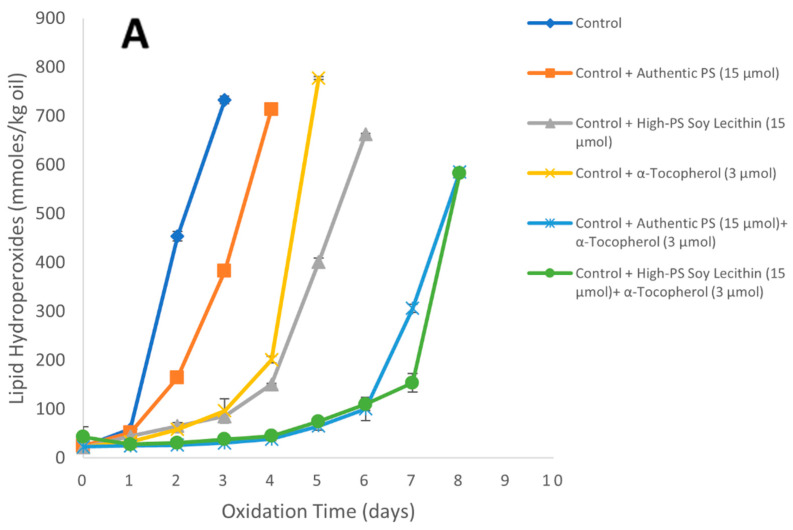
Formation of lipid hydroperoxides (**A**) and hexanal (**B**) in 1% stripped soybean oil-in-water emulsions stabilized with Tween 20 containing 3.0 µmol/kg of α-tocopherol and/or 15.0 µmol/kg of PS or phospholipase-D-modified high-PS soy lecithin at 20 °C. Each value represents the mean (*n* = 3) ± standard deviations. Some error bars lie within the data points.

**Figure 6 foods-11-01014-f006:**
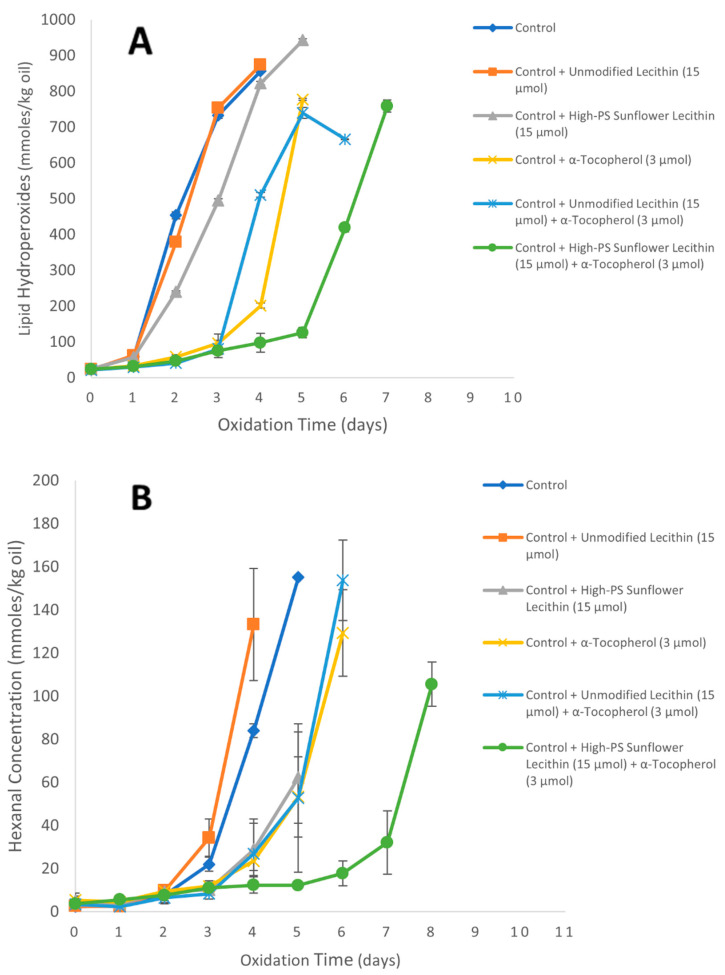
Formation of lipid hydroperoxides (**A**) and hexanal (**B**) in 1% stripped soybean oil-in-water emulsions stabilized with Tween 20 containing 3.0 µmol α-tocopherol/kg of emulsion and/or 15.0 µmol of unmodified or high-PS modified sunflower lecithin/kg emulsion at 20 °C. Each value represents the mean (*n* = 3) ± standard deviations. Some error bars lie within the data points.

**Figure 7 foods-11-01014-f007:**
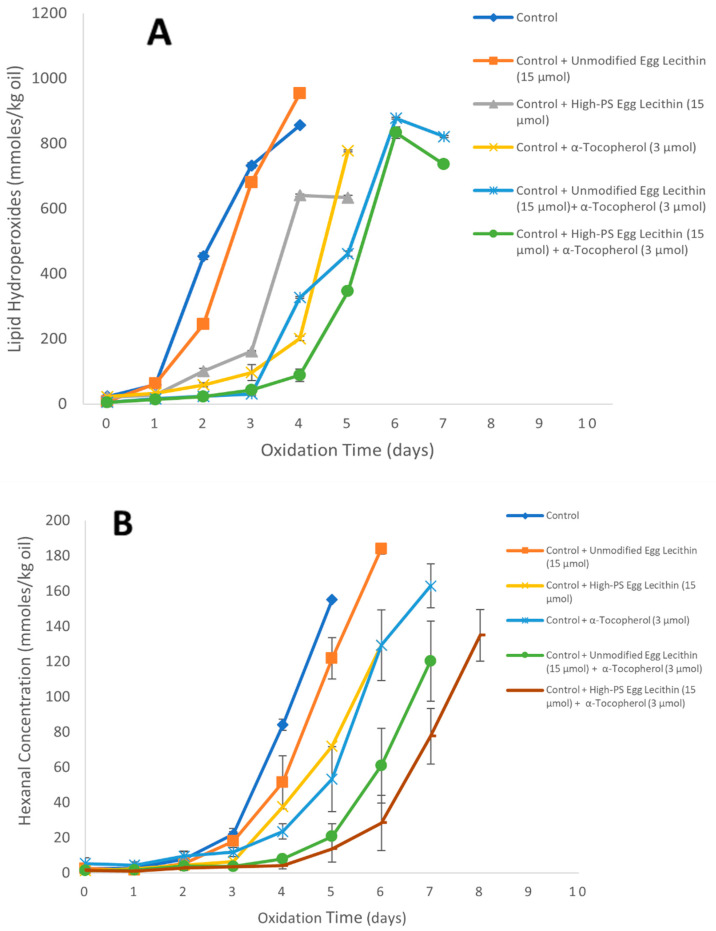
Formation of lipid hydroperoxides (**A**) and hexanal (**B**) in 1% stripped soybean oil-in-water emulsions stabilized with Tween 20 containing 3.0 µmol/kg of α-tocopherol in emulsion and/or 15.0 µmol/kg of unmodified or high-PS egg lecithin in emulsion at 20 °C. Each value represents the mean (*n* = 3) ± standard deviations. Some error bars lie within the data points.

**Table 1 foods-11-01014-t001:** Percentage of different phospholipids found in common plant and animal sources (Table modified from Szuhaj [28]).

Percentage of Total Phospholipids in Lecithins						
Phospholipid	Soybean	Rapeseed	Sunflower	Egg Yolk	Milk	Corn
Phosphatidylcholine	24	25	25	74	27	30
Phosphatidylethanolamine	22	22	11	19	36	3
Phosphatidylinositol	15	15	19	1		16
Phosphatidic acid	7		3			9
Other phospholipids (including phosphatidylserine)	5	19		1	8	

**Table 2 foods-11-01014-t002:** Solvent gradient system used for phospholipids class HPLC separation.

	Gradient System for Phospholipids Class Separation			
Time (min)	Flow Rate (mL/min)	Hexane (%)	Isopropanol (%)	Water (%)
0.01	0.8	53	42	5
5	1	40	55	5
10	1.25	40	50	10
30	1.25	40	55	5
35	0.8	53	42	5

**Table 3 foods-11-01014-t003:** Different parameters analyzed for optimizing the reaction conditions for the conversion of PC from soy lecithin to PS (low, medium, and high describe the PS yields).

Parameter	Conditions				
Enzyme concentration (PLD units)	4 low	8 high	12 high	16 low	20 low
Serine concentration (M)	1 M low	1.5 M low	2 M medium	2.5 M high	3 M high
Acetate buffer content	2 mL low	4 mL medium	6 mL high	8 mL low	10 mL low

## Data Availability

Data is contained within the article.

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
