# Peer review of "Production of a High-Phosphatidylserine Lecithin That Synergistically Inhibits Lipid Oxidation with α-Tocopherol in Oil-in-Water Emulsions"

_foods, 2022, doi:10.3390/foods11071014_

Round 1

Reviewer 1 Report

The manuscript “Production of a High Phosphatidylserine Lecithin Which Synergistically Inhibits Lipid Oxidation with α-Tocopherol in Oil In-Water Emulsions” describes the optimization of conversion of Phosphatidylcholine to Phosphatidylserine in natural lecithins by using Phospholipase D from Streptomyces sp. and the use of the resulting high phosphatidylserine lecithin to protect Oil In-Water Emulsions from peroxidation using the synergistic effect with α-Tocopherol.

The work is well conducted and described.

minor revisions comments:

In different points of the manuscript the Greek letters appear as a spiral symbol.

Always use the upper case L for liters

The atomic indexes in the chemical formulas must be subscript

The temperature is XX °C and not XX°C

Row 22: “increases” instead of “increase”

Row 91: full mark at the end

Row 127: “to determine if…” instead of “to determine is…”

Row 176: provide details of the HPLS system as in row 185

Row 217: provide the pH value with an additional digit (e .g. 7.0)

Row 223: if possible provide the RPM value instead of the generic “low speed”

Row 228: acronym GC never explained before

Row 233: 10 mM (separated)

Row 257: provide details of the gas chromatograph system

Row 351: provide the PDI

Row 408: why the interaction indexes are different from those in row 405 if the lag phases increase are the same?

Row 409: it is not so surprising since the α-tocopherol concentration in PS soy lecithin is 1/10 than that added externally

Row 473: “not be surprising” instead of “not surprising”

Row 489: in conclusion,

Row 489: specify the source of phospholipase D.

Author Response

All changes were made.  Interaction indexes are based on the lag phases of the individual antioxidants and the combined antioxidants.  Therefore, differences in the individual lag phases changed the interaction indexes.

Reviewer 2 Report

The present study shows synergistic activity of enzymatically modified lecithin rich in PS. The data obtained will give great benefit to increase the oxidative stability of unsaturated lipids in food products. In addition, they are also giving new information in the field of lipid chemistry. The followings should be concerned.

Reference list: author names of some references are all uppercase letters. In addition, Journal names are missing in some references. Please correct them.

Lines 239-240: Hydroperoxides measurements were ….. gas chromatography (GC) analysis. What GC?

Tocopherol content of unmodified lecithin should be shown in somewhere.

Lines 402-416: There may be an optimum concentration of tocopherol and PS in their synergistic activity.

3.6: The endogenous tocopherol level of the modified sunflower lecithin should be described. The data may be useful to discuss why the modified soybean lecithin alone could show the higher antioxidant activity even without tocopherol addition.

Author Response

References were correct, sorry but something with Endnote did not work correctly.

Tocopherol concentrations were add in line 485 as well as a comment on that the differences between unmodified soy and sunflower lecithin were not likely due to differences in tocopherol concentrations as they were similar.